# Rethinking the Implicit Optimization Paradigm with Dual Alignments for Referring Remote Sensing Image Segmentation

## ABSTRACT

Referring Remote Sensing Image Segmentation (RRSIS) is a challenging task that aims to identify specific regions in aerial images that are relevant to given textual conditions. Existing methods tend to adopt the paradigm of implicit optimization, utilizing a framework consisting of early cross-modal feature fusion and a fixed convolutional kernel-based predictor, neglecting the inherent inter-domain gap and conducting class-agnostic predictions. In this paper, we rethink the issues with the implicit optimization paradigm and address the RRSIS task from a dual-alignment perspective. Specifically, we prepend the dedicated Dual Alignment Network (DANet), including an explicit alignment strategy and a reliable agent alignment module. The explicit alignment strategy effectively reduces domain discrepancies by narrowing the inter-domain affinity distribution. Meanwhile, the reliable agent alignment module aims to enhance the predictor's multi-modality awareness and alleviate the impact of deceptive noise interference. Extensive experiments on two remote sensing datasets demonstrate the effectiveness of our proposed DANet in achieving superior segmentation performance without introducing additional learnable parameters compared to state-of-the-art methods.

## CCS CONCEPTS

• **Computing methodologies** → **Image segmentation**; *Scene understanding*; Image representations.

## KEYWORDS

remote sensing image, referring image segmentation, transformer

## 1 INTRODUCTION

With the development of deep learning [19, 25], remote sensing image segmentation [17, 32, 38], which provides rich insights into the earth's surface through pre-defined surface categories conditioned on aerial images, has made rapid progress with many applications such as environmental monitoring [7], land cover classification [29], detailed mapping of terrain and urban areas [6], *etc.* However, in practical scenarios, distinct from traditional single-modality segmentation (purely visual remote sensing image segmentation), there is often a need to segment specific regions in aerial images, unrestricted by fixed class labels, but rather guided by text descriptions

*ACM MM, 2024, Melbourne, Australia*

© 2024 Copyright held by the owner/author(s). Publication rights licensed to ACM.
ACM ISBN 978-x-xxxx-xxxx-x/YY/MM
https://doi.org/10.1145/nnnnnnn.nnnnnnn

with the richer vocabulary and syntactic variations inherent in human natural language. To adapt to free-form text conditions, many works have have turned their attention to the referring remote sensing image segmentation (RRSIS) [23, 42]. Since the spatial and geographical differences conveyed from an aerial perspective differ from natural images, how to fully exploit the given text conditions to perform accurate and text-relevant visual region segmentation is thus extremely challenging.

Currently, mainstream RRSIS methods draw inspirations from referring image segmentation (*e.g.*, LAVT [39]) credited to competitive performance, striving to align visual and linguistic branches. The core idea is, as illustrated in Fig. 1 (a), RRSIS methods [23, 42] tend to resort to a paradigm of fusion-then-segmentation, including an implicit alignment between pre-trained textual (*e.g.*, BERT) and visual (*e.g.*, Swim-B) streams, and a fixed convolutional kernel for dense prediction. For example, Yuan *et al.* [42] directly integrate semantic features extracted by BERT into visual feature extraction, enabling semantic guidance for foreground focus, while RMSIN [23] further employs a cross-scale interaction strategy to replace skip connections and simply maps visual features to predictions with a convolutional kernel. Overall, these methods mostly employ ambiguous text-visual feature fusion and utilize semantics-agnostic predictors for pixel-level classification, relying solely on implicit interaction to pray for ideal optimization. These implicit optimization-based methods have made strides in referring remote sensing image segmentation indeed, but struggling to produce ideal results when dealing with complex semantic scenarios.

Despite their promising results, after an in-depth analysis of implicit optimization that exists in the current fusion-then-segmentation paradigm, we find two key ingredients lacking in previous works. **(1) Inter-domain Misalignment.** During the visual-textual interaction process, the pre-trained knowledge of the language encoder originates from the natural language processing (NLP) domain, representing a discrete, structured data format, while the visual encoder focuses solely on the encoding and parsing of natural image domain, tending towards a continuous and high-dimensional representation. This notable gap between two domains, due to the inherent differences in training data, leads to a misalignment of input distributions in existing methods' implicit alignment paradigm (direct language-visual interaction), resulting in ambiguous foreground activation, as shown in Fig. 1 (b) left. Designing an appropriate strategy to *ensure the explicit alignment of visual and textual domain information* is worthy of exploration. **(2) Semantics-agnostic Prediction.** Existing methods [23, 42] employ a CNN-based predictor for dense prediction (Fig. 1 (a)), which remains fixed post-training and struggles to adapt to diverse visual-textual input pairs. Without explicit guidance from textual information, the fixed semantics-agnostic predictor fails to deeply understand objects and scenes in images,

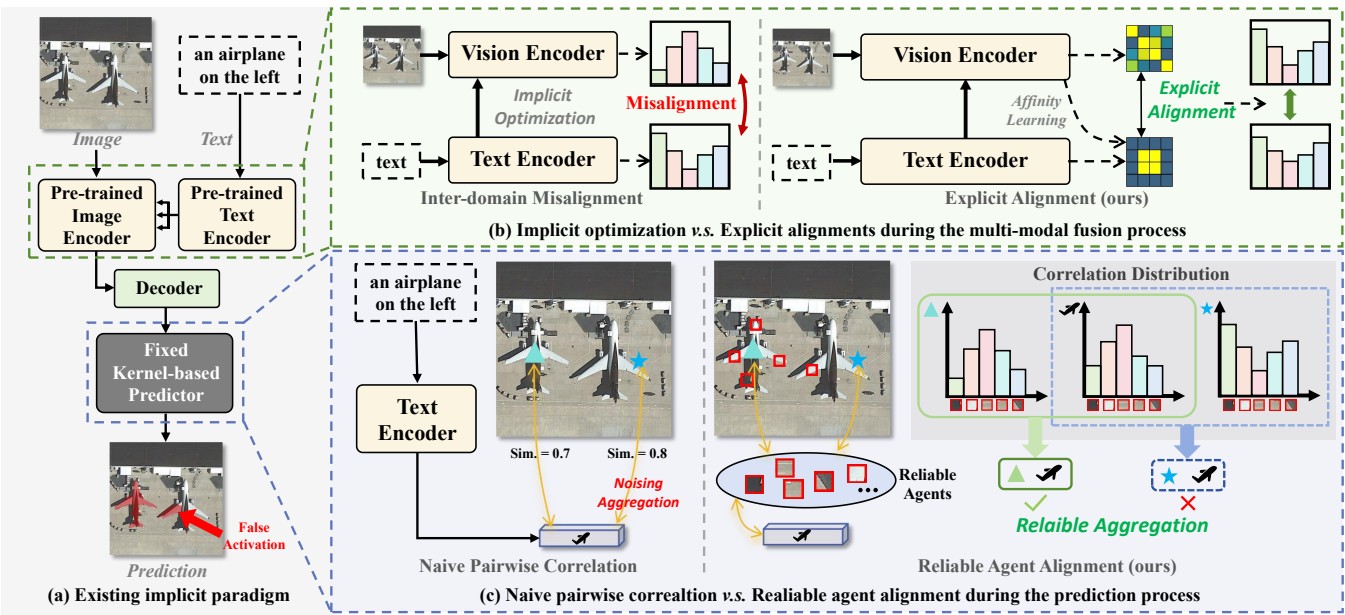

**Figure 1: Illustration of our motivation. (a) shows the the implicit optimization paradigm of existing RRSIS methods, leading to issues of the inter-domain misalignment (*implicit cross-modal fusion*) and class-agnostic predictions (*fixed kernel-based predictor*). (b) shows the comparison of implicit optimization and our explicit alignment during the muti-modal fusion process. (c) shows the comparison of naive pairwise correlation and our reliable agent alignment during the prediction process.**

lacking generalisability across diverse linguistic contexts. Addressing how to *endow the predictor with semantic perceptual capabilities to focus on specific targets* is a desirable question.

In this paper, we argue that pursuit alignments matter in RRSIS, which is intuitively sensible from the definition of the task itself. In this paper, we endeavor to mitigate the adverse effects of implicit optimization (misaligned encoding and semantics-agnostic segmentation), and aim to explore to achieve explicit alignments of visual-textual cues in the RRSIS task. Specifically, we design a coherent **D**ual **A**lignment **Net**work (DANet) for referring remote sensing image segmentation, including an affinity-based explicit alignment strategy and an agent-based reliable alignment module. **In the explicit affinity alignment strategy**, we endeavor to explicitly alleviate the domain gap between vision and language. Intuitively, for the activation of textual information within visual features, pixels identified as foreground should exhibit a high correlation with each other. Drawing inspiration from this instinct, we harness insights from affinity learning [13] to seek perceptibility to spatial position and discrimination to surrounding pixels. We deem that there should be consistency between the affinity of text-to-visual activation and the internal affinity of visual features, as shown in Fig. 1 (b) right. Therefore, the explicit affinity alignment strategy is designed to utilize text-to-visual activation as pseudo-labels, applying affinity constraints between pseudo-labels and corresponding hierarchical visual features to narrow their distributions toward convergence, achieving explicit visual-textual domain alignment across multiple levels without the introduction of additional learnable parameters.

**In the reliable agent alignment module**, our intention is to design a semantics-aware classifier with alignment with both domains. Since fixed classifiers cannot perceive semantic information post-training, a naive idea is to utilize pairwise alignment to update textual embeddings to become classifiers, as depicted in Fig. 1 (c) left. However, in remote sensing scenarios, directly applying pixel-sentence interaction inevitably increases the risk of unreliability due to the deceptive noise caused by long-distance imaging patterns in aerial imagery. In this case, segmentation networks are naturally equivocal for similar vision clues, leading to sub-optimal results. Hence, we aim to construct robust interaction with reliable matching to avoid uncertainty interference. Intuitively, as shown in Fig. 1 (c) right, for the similarity distribution between a visual area and a group of pixels, the current distribution of similarity between the corresponding semantics and the same group should appear similar, and vice versa. Inspired by this intuition, we adaptively select reliable agents from vision features, avoiding deceptive visual interference caused by direct matching. In this way, we are able to leverage the purified textual knowledge to channel its attention towards discerning and prioritizing foreground regions. Overall, we manage to address problems of the implicit optimization in RRSIS from a dual-alignment perspective, leveraging a comprehensive framework that combines explicit affinity alignment and reliable agent alignment strategies. Through this integrated approach, we not only mitigate the challenges posed by domain discrepancies but also equip the predictor with semantic awareness, thereby enhancing its ability to discern and prioritize foreground regions accurately.

In this work, our contributions can be summarized as follows:

- We analyze the issues within the implicit optimization paradigm of referring remote sensing image segmentation, and propose to mitigate domain discrepancies from an explicit alignment perspective.
- Specifically, we design a Dual-Alignment Network (DANet) including an explicit affinity alignment strategy to alleviate the inter-domain gap and a reliable agent alignment module to equip the predictor with semantic awareness.
- Extensive experimental results on multiple challenging benchmarks demonstrate that our proposed method performs favorably against state-of-the-art referring remote sensing image segmentation methods.

## 2 RELATED WORK

### 2.1 Remote Sensing Image Segmentation

Remote sensing image segmentation is a crucial task in various scientific and engineering applications, facilitating the extraction of valuable information from satellite or aerial images [17, 32, 38]. Early researches on remote sensing image segmentation rely heavily on handcrafted features and simple thresholding methods [3, 26, 27, 36], often struggle with complex scenes and lacked robustness to variations in illumination and terrain. With the advent of deep learning, there has been a paradigm shift towards data-driven approaches that learn hierarchical representations directly from the input data. CNN-based works [1, 2, 33] adopt well-known architectures like VGG [31], ResNet [9], and U-Net [30], demonstrating superior performance in extracting spatial features from remote sensing images. Transformer-based methods [8, 35, 38] introduce the ability to model long-range dependencies, further enhancing the remote sensing scenes perception.

However, traditional methods for remote sensing image segmentation often struggle to meet the specific foreground recognition requirements in practical applications. Therefore, referring remote sensing image segmentation (RRSIS) [23, 42] guided by textual cues has emerged. In specific, Yuan et al. [42] introduce a dataset for the RRSIS task and employed fine-grained textual information to guide visual encoding layer by layer. RMSIN [23] further incorporates a cross-scale interaction module to integrate multi-scale information from the encoders and used a fixed convolutional kernel to map the final layer visual features to predictions. Nevertheless, these existing methods mostly employ ambiguous text-visual feature fusion and utilize semantics-agnostic predictors for prediction, relying solely on implicit interaction to pray for ideal optimization. This implicit optimization paradigm ignores the inherent differences across domains and fails to mine valuable clues dealing with complex semantic scenarios. In this paper, we strive to address issues of the implicit optimization paradigm from an dual alignment perspective.

### 2.2 Referring Image Segmentation

Referring Image Segmentation (RIS), aiming to segment specific objects within an image based on natural language expressions [15, 21, 41], has garnered significant attention in recent years. The task's goal is to understand and interpret linguistic descriptions, such as "the red phone" or "the plane on the right", and precisely locate and segment the referred objects in the image. Early RIS

research [20, 40] focuses on extracting visual and linguistic features separately with Convolutional Neural Networks (CNNs) [18] and Long Short-Term Memory (LSTM) [10] networks, struggling to capture the relationships between language expressions and visual contents. The recent emergence of Transformer [34] architectures revolutionize RIS methodologies, offering remarkable fusion capabilities for multi-modality integration. For example, MDETR [16] showcases the effectiveness of simple concatenation of vision and language features followed by Transformer encoding and decoding for various vision-language tasks. Building upon this, VLT [5] pioneers the integration of Transformer architecture into referring segmentation, employing query generation modules to enrich language expressions with contextual image information. To facilitate cross-modal integration, LAVT [39] introduces language-aware attention mechanisms into image encoding processes, aiding early fusion of cross-modal features and improving segmentation accuracy. However, these existing referring image segmentation works primarily focus on the understanding of natural image domains, making it challenging to address complex terrains and specific contexts in aerial remotely sensed scenes. Thus, Yuan et al. [42] propose language-guided cross-scale enhancement to encourage the exploration of semantic clues in aerial images, while RMSIN [23] further introduces cross-scale skip connections to boost performance. Despite these efforts, these methods often fall into the paradigm of implicit optimization, neglecting the inter-domain gap and conducting semantic-agnostic prediction. We rethink this implicit optimization paradigm with dual alignments to better interpret referring remote sensing image scenes.

## 3 METHOD

In this section, we first present the overview of the proposed DANet for RRSIS and the baseline model in Sec. 3.1. Then, we describe the details of the explicit affinity alignment strategy in Sec. 3.2 and the reliable agent alignment module in Sec. 3.3. Finally, in Sec. 3.4, the training and inference procedure are discussed.

### 3.1 Overview

As shown in Fig. 2, the input pair of referring remote sensing image segmentation contains an input image $\mathbf{I} \in \mathbb{R}^{H \times W \times 3}$ and an input expression $\mathbf{T} \in \mathbb{R}^L$, where $H$ and $W$ refer to the height and width the input image $\mathbf{I}$, $L$ denotes the number of words. For the feature extraction, we utilize the pre-trained language encoder (e.g., BERT) for encoding linguistic information as fine-grained embeddings $\mathbf{L}_c \in \mathbb{R}^{L \times C_t}$ and global textual representation $\mathbf{L}_g \in \mathbb{R}^{1 \times C}$, and take LAVT [39] as our baseline model, including Swin Transformer [24] as our backbone for vision feature extraction and PWAM (pixel-word attention module, which is based on cross-attention) for vision-language fusion. During Stage $i$ in the encoder, the input feature $\mathbf{F}_{i-1}$ is fused with text features $\mathbf{L}_c$ to output enriched vision feature $\mathbf{F}_i$ after downsampling. Please note that this part is not the focus of our design concerns, we use the same settings as LAVT for fair comparison.

To mitigate the inter-domain gap caused by different pre-trained knowledge of vision and language encoders, the extracted hierarchical vision embeddings $\{\mathbf{F}_l\}_{l=2}^4$ are utilized by the explicit affinity alignment strategy to realize explicit alignments between vision

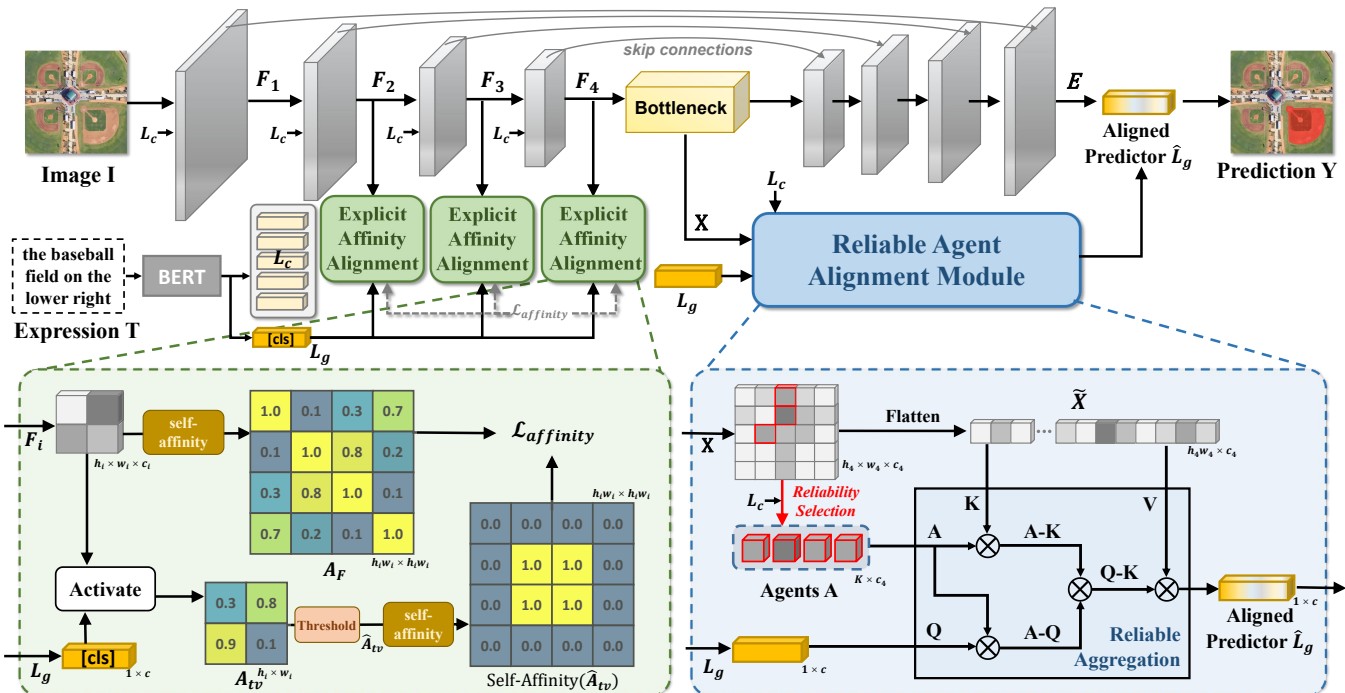

**Figure 2: Framework of our proposed DANet. It includes an explicit affinity alignment strategy (Sec. 3.2) to hierarchically narrow the inter-domain affinity distribution and a reliable agent alignment module (Sec. 3.3) to acquire an adaptive predictor aware of both modalities meanwhile mitigate the noising aggregation.**

and language domains. Through this operation of narrowing affinity distributions, the activation of global textual representation $\mathbf{L}_g$ in vision features gradually focuses on the ideal region. To empower the classifier's vision-aware capacity and further reduce deceptive background interference, $\mathbf{L}_g$ is regarded as a semantics-initialized prototype to align with selected reliable agents $A$ to acquire aligned predictor $\hat{\mathbf{L}}_g$ in the reliable agent alignment module. Finally, the top-level feature $\mathbf{E}$ from the backbone decoder is interacted with the evolved reliable predictor $\hat{\mathbf{L}}_g$ to generate the prediction $\mathbf{Y}$.

### 3.2 Explicit Affinity Alignment Strategy

To tackle the problem of inter-domain gap caused by different pre-trained knowledge in vision and language, we carefully design an explicit alignment strategy to narrow the domain discrepancies. Drawing inspiration from affinity learning, we explore explicitly aligning the vision and language domains by approximating affinity distributions. It can be noted that, in addition to leveraging the fine-grained textual features $\mathbf{L}_c \in \mathbb{R}^{L \times C_t}$, our design also strategically retains the [CLS] token $\mathbf{L}_g \in \mathbb{R}^{1 \times C}$ from BERT outputs, serving as a macroscopic representation of global textual features. The [CLS] token in BERT serves a significant advantage: it encapsulates the semantic understanding of the entire input sequence, acting as a comprehensive sentence-level embedding that captures contextual information efficiently.

Formally, during Stage $i$ in the backbone encoder, for the vision feature $\mathbf{F}_i \in \mathbb{R}^{h_i \times w_i \times c_i}$, we first flatten $\mathbf{F}_i$ as $\tilde{\mathbf{F}}_i \in \mathbb{R}^{h_i w_i \times c_i}$, and the

self-generated pixel-level affinity from $\tilde{\mathbf{F}}_i$ is formulated as:

$$\text{Self-Affinity}(\tilde{\mathbf{F}}_i) = \text{Softmax}\left(\frac{\tilde{\mathbf{F}}_i \cdot \tilde{\mathbf{F}}_i^\top}{\|\tilde{\mathbf{F}}_i\|^2}\right), \quad (1)$$

where $\top$ refers to the transpose operation. Here, we denote the self-affinity map as $\mathbf{A}_F$ for clarity. The internal self-affinity of image features represents their internal consistency or self-similarity, each value in the self-affinity map reflects the correlation between the current observation area and other pixels within the same domain, indicating how well the features at different locations align with each other. To explicitly align the vision affinity with the language domain, we further utilize the global textual representation $\mathbf{L}_g$ to activate the corresponding semantic clues in vision feature $\mathbf{F}_i$. $\mathbf{F}_i$ is mapped into the same dimension with $\mathbf{L}_g$ and produce the text-to-visual activation $\mathbf{A}_{tv}$. To convert the multi-modal activation $\mathbf{A}_{tv}$ into hard labels, we conduct the transformation based on thresholding:

$$\hat{\mathbf{A}}_{tv} = \begin{cases} 1, & \text{if } a(x,y) > \alpha \\ 0, & \text{otherwise} \end{cases} \quad (2)$$

where $a(x,y)$ is the original pixel value from the activation map, and $\alpha$ is the threshold for filtering positive samples. During implementation, we set $\alpha$ as 0.80 by default, and as shown in Tab. 6, the variation of $\alpha$ has insignificant impacts on performance, indicating the robustness of our design. Then, we aim to realize inter-domain alignment by narrowing the affinity distribution between vision and language clues. Specifically, we apply affinity constraints between these pseudo-labels and corresponding hierarchical visual

features to narrow their distributions towards convergence, which can be formulated as:

$$\mathcal{L}_{affinity} = \|\mathbf{A}_F \odot \text{Self-Affinity}(\hat{\mathbf{A}}_{tv}) - \text{Self-Affinity}(\hat{\mathbf{A}}_{tv})\|_F^2, \quad (3)$$

where $\|\cdot\|_F$ denotes the Frobenius norm, encouraging the distance between the two affinities to be similar and aligned. $\odot$ denotes the element-wise multiplication, which signifies that we only focus on affinity constraints within the target, without enforcing consistency in the distribution of different categories in the background.

One notable advantage of this explicit alignment strategy is that it operates within the text-visual encoding process without adding new learnable parameters, achieving gains in segmentation performance. This ensures generalization and scalability across multiple scenes, facilitating explicit visual-textual domain alignment in a principled and efficient manner.

## 3.3 Reliable Agent Alignment Module

Existing approaches mostly use fixed semantics-agnostic predictors for dense prediction, struggling to mine valuable clues when facing complex semantic scenarios. In order to make the predictor semantic-aware and aligned to the linguistic domain, a preliminary scheme is to use global textual representations $\mathbf{L}_g$ as prototype, and directly align with the semantically-rich visual features $\mathbf{X}$ after the bottleneck to obtain a visual textually multi-aware classifier. However, such a naive correlation would inevitably fall into the trap of deceptive noise in remote sensing imagery, as shown in Fig. 1 (c) left, destroying the purity of semantic features. How to achieve reliable visual-text alignment and avoid noise interference simultaneously is a problem worth solving. Intuitively, in a situation where it is difficult to distinguish between pixel classes, humans tend to decide by looking for a comparison between the current query location and a reliable class-aware region. Motivated by this intuition, assuming that given a trusted medium in remotely sensed imagery, we argue that the distribution of semantic and foreground pixels over that medium should be consistent, and vice versa. Thus we design a novel reliable agent alignment module (RAAM) to find dependable points as reliable medium (*i.e.*, agent) to acquire more accurate correlations.

Given the bottom-level image feature $\mathbf{X}$ after bottleneck ($\mathbf{F}_4$ through feature enhancement) derived from the backbone encoder, the queries arise from the global textual representation $\mathbf{L}_c = \{l_n\}_{n=1}^L$, and keys and values arise from the input features $\tilde{\mathbf{X}} = [f_1; f_2; ...; f_{hw}]$ (flattened $\mathbf{X}$). Formally,

$$\mathbf{Q}_n = l_n \mathbf{W}^Q, \mathbf{K}_m = f_m \mathbf{W}^K, \mathbf{V}_m = f_m \mathbf{W}^V, \quad (4)$$

where $n \in [1, ..., L]$, $m \in 1, 2, ..., hw$ and $\mathbf{W}^Q \in \mathbb{R}^{C_t \times C_k}$, $\mathbf{W}^K \in \mathbb{R}^{C \times C_k}$, $\mathbf{W}^V \in \mathbb{R}^{C \times C_v}$ are linear projections. Then, we can obtain the correlation between queries and keys with distance calculation as:

$$corr_{n,m} = \frac{dis(\mathbf{Q}_n, \mathbf{K}_m)}{\sqrt{C_k}}, \quad (5)$$

where $dis(\cdot, \cdot)$ denotes the distance metric.

Direct pairwise alignment is unreliable due to deceptive backgrounds during remote sensing, especially with similar texture details, *e.g.*, bikeways and sidewalks. The reliability for each pixel

can be obtained via the weighted sum over all correlations as:

$$r_m = \sum_{n=1}^L corr_{n,m}, m \in 1, 2, ..., hw, \quad (6)$$

where the top-$K$ pixels are selected with the highest reliabilities (*i.e.*, the largest correlations with semantics) to be agents $\mathbf{A}$. The corresponding features can be denoted as $\mathbf{F}^A = \{f_k^A\}_{k=1}^K$. With agents that filter fine-grained semantic activations, we use them as mediators to achieve reliable alignment between visual features $\mathbf{X}$ and global textual representations $\mathbf{L}_g$. We first map $\mathbf{L}_g$ to the query dimension (*i.e.*, the same dimension with $\mathbf{K}_m$) as $\mathbf{Q}^g = \mathbf{L}_g \mathbf{W}^{Q^g}$, where $\mathbf{W}^{Q^g} \in \mathbb{R}^{C \times C_k}$. Then, we calculate the agent-global semantics correlation $corr_n^s$ and the agent-pixel correlation $corr_m^p$ respectively as same as Eq.(5):

$$corr^g = \text{softmax}(\frac{\mathbf{Q}^g(\mathbf{F}^A \mathbf{W}^K)^\top}{\sqrt{C_k}}),$$
$$corr_m^p = \text{softmax}(\frac{(f_m \mathbf{W}^Q)(\mathbf{F}^A \mathbf{W}^K)^\top}{\sqrt{C_k}}). \quad (7)$$

And then, we can obtain the reliable alignment between global semantics and pixels based on agents $\mathbf{A}$ as:

$$align_m^{g\&p} = corr^g(corr_m^p)^\top, \quad (8)$$

which is used to acquire more accurate relations. Finally, the updated aligned classifier (*i.e.*, clear vision-aware semantic representations) can be acquired by blending values with the reliable alignment $align_m^{g\&p}$ as:

$$\hat{\mathbf{L}}_g = \sum_{m=1}^{hw} align_m^{g\&p} \mathbf{V}_m, \quad (9)$$

and following general transformer pipeline [34], we equip the reliable classifier $\tilde{\mathbf{L}}_g$ with self-attention and FFN at the output of the reliable agent alignment module. In this way, the global semantics $\mathbf{L}_g$ is modified in the RAAM and finally evolves into a reliable aligned predictor $\hat{\mathbf{L}}_g$.

## 3.4 Training and Inference

With the final high-resolution features $\mathbf{E} \in \mathbb{R}^{H \times W \times C}$ from the backbone upsampling decoder and the aligned predictor $\hat{\mathbf{L}}_g \in \mathbb{R}^{1 \times C}$, we can finally obtain the segmentation map $\mathbf{Y}$ as:

$$\mathbf{Y} = \mathbf{E} \times \hat{\mathbf{L}}_g^\top. \quad (10)$$

For better training our network, we utilize the affinity constraint within the designed explicit alignment strategy and the conventionally used loss paradigm [23, 39, 42], *i.e.*, the binary cross-entropy loss for mask recognition. The final loss is formulated as:

$$\mathcal{L}_{total} = \mathcal{L}_{BCE}(\overline{\mathbf{Y}}, \mathbf{Y}) + \lambda_{af} \mathcal{L}_{affinity} \quad (11)$$

where $\overline{\mathbf{Y}}$ is the ground-truth of the prediction $\mathbf{Y}$, $\lambda_{af}$ denotes the coefficient of $\mathcal{L}_{affinity}$.

**Table 1: Comparisons of existing referring remote sensing image segmentation methods on the RefSegRS [42] dataset. The best results are shown in bold. † denotes the performance of our reproduction of the corresponding model.**

| Method | Visual Enc. | Text Enc. | mIoU | oIoU | Pr@0.9 | Pr@0.8 | Pr@0.7 | Pr@0.6 | Pr@0.5 |
|---|---|---|---|---|---|---|---|---|---|
| RRN [20] | ResNet-101 | LSTM | 43.34 | 66.12 | 1.10 | 7.59 | 15.30 | 23.39 | 31.21 |
| CMSA [41] | ResNet-101 | None | 41.47 | 64.53 | 0.83 | 5.61 | 12.71 | 20.25 | 28.07 |
| LSCM [14] | ResNet-101 | LSTM | 38.64 | 63.21 | 1.23 | 6.12 | 10.53 | 21.56 | 32.12 |
| CMPC [12] | ResNet-101 | LSTM | 33.57 | 61.25 | 0.88 | 8.94 | 11.26 | 16.34 | 26.57 |
| BRINet [11] | ResNet-101 | LSTM | 32.87 | 60.16 | 1.27 | 3.52 | 10.11 | 15.74 | 22.56 |
| CMPC+ [22] | ResNet-101 | LSTM | 54.21 | 68.23 | 3.27 | 12.56 | 29.54 | 47.65 | 51.27 |
| LAVT [39] | Swin-B | BERT | 57.74 | 76.46 | 4.51 | 15.41 | 32.14 | 57.40 | 71.44 |
| RMSIN† [23] | Swin-B | BERT | 59.63 | 76.29 | 5.38 | 15.89 | 39.37 | 62.83 | 72.26 |
| LGCE [42] | Swin-B | BERT | 59.96 | 76.81 | 5.45 | 16.02 | 39.46 | 61.14 | 73.75 |
| **DANet (ours)** | Swin-B | BERT | **62.14** | **79.53** | **8.04** | **18.29** | **42.72** | **64.59** | **76.61** |

**Table 2: Comparisons of existing referring remote sensing image segmentation methods on the RRSIS-D [23] dataset. The best results are shown in bold. † denotes the performance of our reproduction of the corresponding model.**

| Method | Visual Enc. | Text Enc. | mIoU | oIoU | Pr@0.9 | Pr@0.8 | Pr@0.7 | Pr@0.6 | Pr@0.5 |
|---|---|---|---|---|---|---|---|---|---|
| RRN [20] | ResNet-101 | LSTM | 46.06 | 66.53 | 6.14 | 20.80 | 33.04 | 42.47 | 51.09 |
| CMSA [41] | ResNet-101 | None | 48.85 | 69.68 | 9.02 | 26.55 | 38.27 | 48.04 | 55.68 |
| LSCM† [14] | ResNet-101 | LSTM | 51.35 | 69.28 | 7.93 | 26.37 | 37.87 | 48.04 | 57.12 |
| CMPC† [12] | ResNet-101 | LSTM | 50.41 | 70.15 | 9.31 | 25.28 | 38.50 | 48.85 | 57.93 |
| BRINet [11] | ResNet-101 | LSTM | 51.14 | 70.73 | 9.19 | 28.21 | 39.65 | 49.54 | 58.79 |
| CMPC+† [22] | ResNet-101 | LSTM | 51.41 | 70.14 | 8.16 | 25.91 | 38.67 | 59.36 | 59.19 |
| LAVT [39] | Swin-B | BERT | 61.46 | 77.59 | 24.25 | 43.97 | 53.16 | 63.51 | 69.54 |
| LGCE† [42] | Swin-B | BERT | 61.63 | 77.82 | 23.29 | 45.93 | 55.83 | 66.59 | 71.58 |
| RMSIN† [23] | Swin-B | BERT | 61.96 | 77.63 | 24.71 | 42.30 | 56.01 | 66.87 | **73.87** |
| **DANet (ours)** | Swin-B | BERT | **66.07** | **79.85** | **27.05** | **47.79** | **57.92** | **69.17** | 73.69 |

## 4  EXPERIMENTS

In this section, we will first introduce the datasets used in our work in Sec. 4.1. The implementation details are shown in Sec. 4.2. In Sec. 4.3, we illustrate the specific metric for better evaluation of our method. Then, we further analyze the main results including quantitative evaluations and qualitative results in Sec. 4.4 Finally, we ablate the effectiveness of our method in Sec. 4.5 for better demonstration of DANet's superiority.

### 4.1  Dataset

To demonstrate the effectiveness of our proposed model, we conduct extensive experiments on two referring remote sensing image segmentation benchmarks: RefSegRS [42] and RRSIS-D [23].
**RefSegRS** is a dataset containing 4,420 remote sensing images with language expressions and corresponding labels, divided with 2172 images allocated for training, 431 images for validation, and 1817 images for testing. The image resolution of this dataset is $512 \times 512$.
**RRSIS-D** is a dataset with imaging resolution ranging from 0.5 to 30 meters and a resolution of $800 \times 800$. It consists of 17,402 aerial remote sensing images, with 1,740 images used for testing and the rest for training.

### 4.2  Implementation Details

We adopt Pytorch [28] and Detectron2 [37] to implement the proposed method. 4 NVIDIA GeForce RTX 3090 GPUs are used for training. We take the input image size as $480 \times 480$ following conventional settings [23, 42]. During the training stage, our model is trained with a batch size of 16, using the AdamW optimizer with an initial learning rate of 0.0005. We set the coefficient $\lambda_{af}$ of $\mathcal{L}_{affinity}$ as 0.5, and the number of agents as $K = 32$. We ablate the effects of these hyper-parameters in detail in our ablation studies (Sec. 4.5). Please refer to the supplementary material for more descriptions of implementation details and results.

### 4.3  Metric

For a fair comparison, we adopt the same metric with previous works [23, 42], including Mean Intersection over Union (mIoU), Object Intersection over Union (oIoU), and Precision at X pixels (Pr@X). The higher the values of these metrics, the better the performance. mIoU measures the average overlap between the predicted segmentation masks and the ground truth masks for each class. oIoU focuses on evaluating the segmentation accuracy at the object level rather than at the pixel level. It computes the IoU for each object instance and then calculates the average IoU across

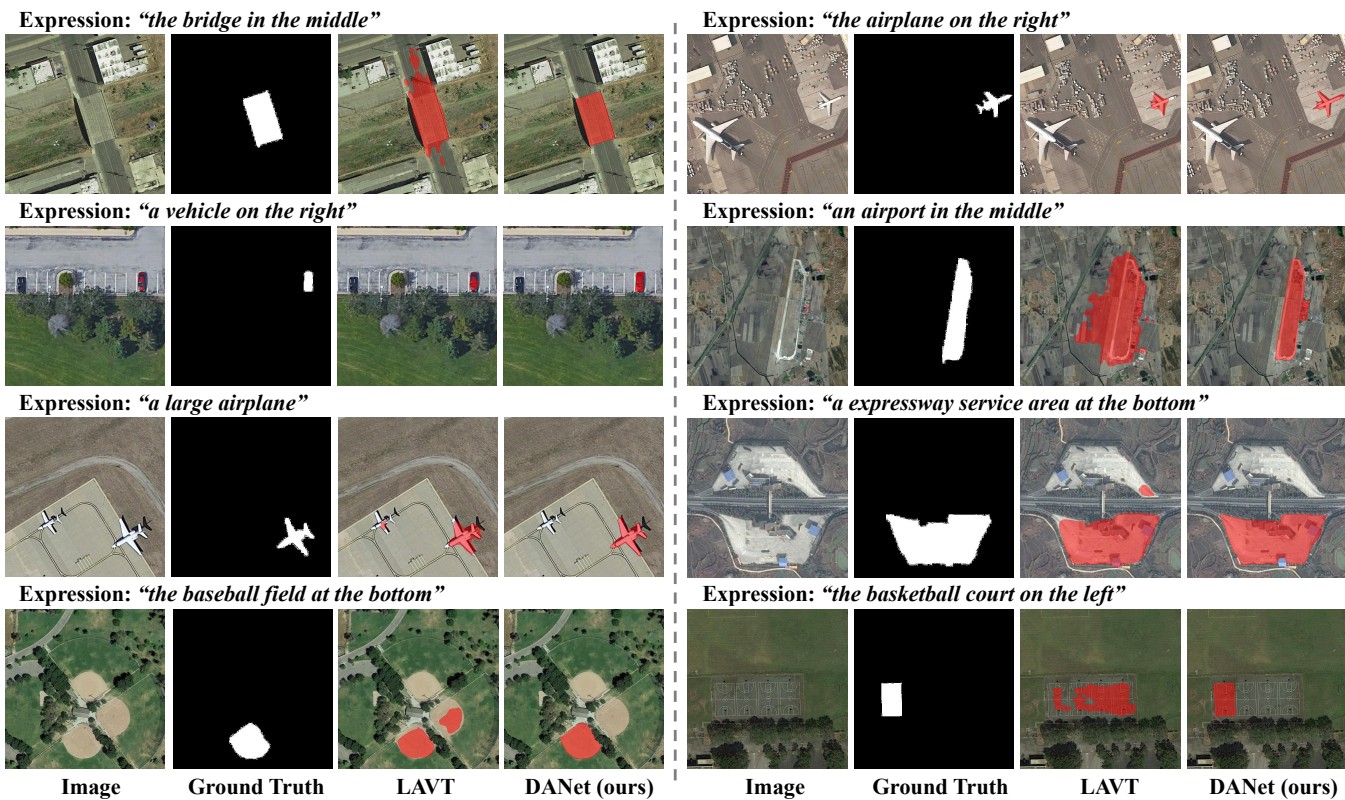

**Figure 3: Qualitative demonstrations of different methods on the RRSIS-D [23] dataset.**

**Table 3: Ablation on main components in terms of mIoU.**

| Main Components | | RefSegRS [42] | RRSIS-D [23] |
|---|---|---|---|
| Affinity Alignment | Reliable Agent Alignment | | |
| ✗ | ✗ | 57.74 | 61.46 |
| ✓ | ✗ | 59.84 | 63.57 |
| ✗ | ✓ | 59.95 | 64.02 |
| ✓ | ✓ | **62.14** | **66.07** |

**Table 4: Ablation on different affinity alignment designs.**

| Hierarchical Affinity | RefSegRS [42] | RRSIS-D [23] |
|---|---|---|
| $\{F_4\}$ | 60.19 | 64.32 |
| $\{F_3, F_4\}$ | 61.88 | 65.73 |
| $\{F_2, F_3, F_4\}$ | **62.14** | **66.07** |
| $\{F_1, F_2, F_3, F_4\}$ | 62.12 | 65.94 |

all objects. Pr@X evaluates the precision of object segmentation by considering the number of correctly predicted pixels within a certain distance threshold X from the ground truth masks.

## 4.4 Main Results

*4.4.1 Quantitative Evaluations.* Our method demonstrates superior performance in referring remote sensing image segmentation, outperforming state-of-the-art methods, as illustrated in Tab. 1 and 2. Evaluation on both RefSegRS [42] and RRSIS-D [23] datasets reveals compelling results. It can be observed that our method not only exhibits superior performance in terms of mIoU and oIoU, but also demonstrates substantial improvements when more precise and fine-grained segmentation is required (*i.e.*, Pr@0.9 and Pr@0.8). Despite challenges posed by the implicit optimization paradigm in remotely sensed scenes, the explicit affinity alignment strategy and reliable agent alignment module in DANet enable accurate identification of foreground regions, mitigating deceptive environmental

interference and the inter-domain gap. Activation maps shown in Fig. 5 further validate the superiority of our alignment design.

*4.4.2 Qualitative Results.* As shown in Fig. 3, DANet shows promising segmentation performance in diverse remotely sensed scenes. In specific, our method performs great in most scenarios for different targets. As shown in the third row in Fig. 3, our approach allows for the accurate discrimination of confusable regions, *e.g.*, as shown in the left of the $4^{th}$ row, DANet isn't tricked by a similar "*baseball field*" to produce activation that shouldn't be there. And in the $2^{nd}$ row, we can observe DANet achieves a high level of completeness in predicting foreground regions, consistent with our findings in Pr@X in Tab. 1 and Tab. 2. With explicit and reliable alignments, DANet can achieve more precise target localisation recognition. As shown in Fig. 5, the activation maps with a reliable alignment strategy demonstrate stronger focusing capability, indicating that our

**Table 5: Comparison of different alignment mechanisms in the reliable agent alignment module.**

| Alignment Mechanism | RefSegRS [42] | RRSIS-D [23] |
|---|---|---|
| cross-attention [34] | 60.17 | 63.92 |
| masked attention [4] | 61.45 | 64.73 |
| **Reliable Agent Alignment** | **62.14** | **66.07** |

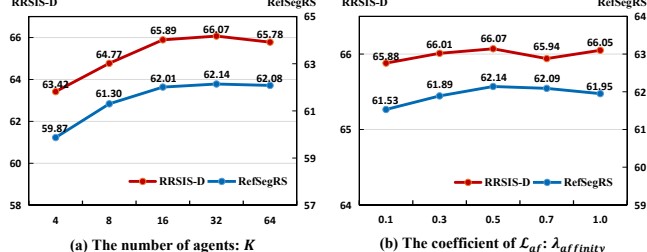

(a) The number of agents: $K$

(b) The coefficient of $\mathcal{L}_{af}$: $\lambda_{affinity}$

**Figure 4: Comparisons of performance with different numbers of agents $K$ and $\lambda_{af}$ in terms of mIoU.**

method can fully leverage language guidance and alleviate domain discrepancies.

## 4.5 Ablation Study

We conduct comprehensive ablation studies on both datasets in terms of mIoU to verify the effectiveness of our modules.

**Effectiveness of the Explicit Affinity Alignment Strategy (EAAS).** As shown in the ablation experiments on the main components of our DANet in Tab. 3, the introduction of affinity explicit alignment yields a discernible performance improvement, elevating the average DSC from 57.74 to 59.84 on the RefSegRS [42] dataset. Besides, the effects of the threshold $\alpha$ in EAAS and the loss coefficient $\lambda_{af}$ are shown in Tab. 6 and Fig. 4 (b), demonstrating our DANet's effectiveness and robustness. EAAS is able to mitigate the inherent inter-domain gap between vision and language, utilizing the affinity constraints to narrow the domain discrepancies. The explicit alignment strategy not only enhances the perception of visual semantic clues but also avoids introducing any additional parameters.

**Table 6: Effects of $\alpha$.**

| $\alpha$ | RefSegRS [42] | RRSIS-D [23] |
|---|---|---|
| 0.75 | 62.03 | 65.94 |
| **0.80** | **62.14** | **66.07** |
| 0.85 | 61.97 | 65.81 |
| 0.90 | 61.63 | 65.72 |

**Impacts about Different Hierarchical Affinity Designs.** As shown in Tab. 4, substantial performance improvements are evident on both datasets when employing multi-level features compared to utilizing only the bottom embedding $F_4$. This underscores the effectiveness of our hierarchical affinity alignment design. The rationale behind this success is that vision-language alignments at different levels are well-suited for capturing targets at diverse scales. However, when applying the top-level feature $F_1$ at the start of EAAS, there exists a drop in performance. We deem that this setback is due to relatively weak ability to capture overall semantic information in $F_1$, thus forcing it to align with semantics may disrupt the initial

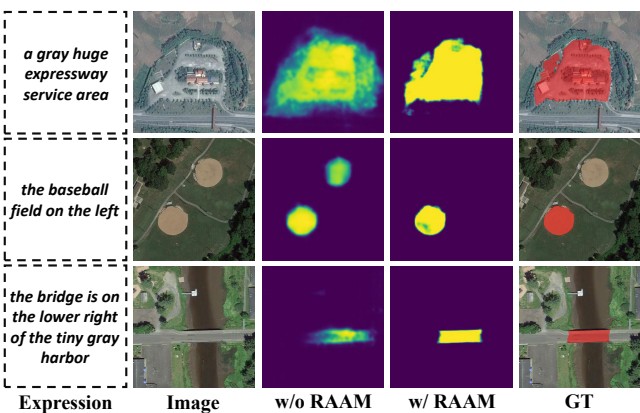

**Figure 5: Visualizations of activation maps w/ and w/o RAAM.**

structure of visual encoding features. The incorporation of suitable hierarchical affinity alignments proves instrumental in effectively parsing various types of targets in remotely sensed scenes.

**Effectiveness of the Reliable Agent Alignment Module (RAAM).** The design of our reliable agent alignment proves immensely beneficial to the model. This is particularly evident when it is integrated with explicit affinity alignments, as indicated in Tab. 3. In Tab. 5, we utilize different mechanisms to interact between vision features $\mathbf{X}$ and the textual representation $\mathbf{L}_g$. Specific benefits brought by reliable agent alignments are demonstrated in Fig. 5, as can be observed, activation without reliable alignment may be subject to deceptive background interference, resulting in erroneous activation or the loss of foreground integrity. Our designed reliable agent alignment module can suppress noise interference, leading to better aggregation and clearer foreground activation of various targets.

**Impacts about the Agent Selection Strategy in RAAM.** In the reliable agent alignment module, the number of agents, denoted as $K$, determines how many reliable pixels are selected to establish the correlation between semantics and pixels. Ablating the number of reference points, as depicted in Fig. 4 (a), reveals that performance reaches its top when $K = 32$, signifying that this number is sufficient for achieving the necessary correction. Too few reliable points may result in insufficient reference samples, leading to an unreliable medium between semantics and pixels, while too many agents may overly emphasize model discrimination and include irrelevant background information, ultimately disrupting the correlation distribution. The adaptive selection of agents in RAAM holds pivotal importance for pixel-level precision in challenging regions.

## 5 CONCLUSION

In this paper, we rethink the implicit optimization paradigm to address the RRSIS task from a dual-alignment perspective. Specifically, we design a Dual Alignment Network (DANet) including an explicit alignment strategy and a reliable agent alignment module. The explicit alignment strategy effectively reduces domain discrepancies by narrowing the inter-domain affinity distribution, and the reliable agent alignment module enhances the predictor's multi-modality awareness and alleviates the impact of deceptive noise interference. Extensive experiments demonstrate our DANet's effectiveness.

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
