# OpenReview forum: "Rethinking the Implicit Optimization Paradigm with Dual Alignments for Referring Remote Sensing Image Segmentation"
_acmmm.org/ACMMM/2024/Conference — MM2024 Poster_

### Official Review · Reviewer_Svcm · 2024-05-16

**Rating:** 4
**Confidence:** 3

**Summary:**

This paper prepend the dedicated Dual Alignment Network (DANet) for RRSIS. Extensive experiments on two remote sensing datasets demonstrate the effectiveness of the proposed DANet in achieving superior segmentation performance without introducing additional learnable parameters compared to state-of-the-art methods.

**Strengths:**

1.This paper proposed a noval framework for RRSIS and has clear idea.
2.The experimental results are good and the comparison methods are complete.

**Limitations:**

- 1.In Figure 3, almost all of them are large targets, and the questions asked are very simple, all about location and size. RRSIS-D has many cases of small targets, which are not visualized.
- 2.What are the application requirements of RRIS in remote sensing? For example, in the first line of Figure 3, there is only one bridge in the image, so what is the role of the text description. Even if text is not provided, the existing semantic segmentation model can extract the bridge very well.
- 3.The SAM model can also provide text descriptions as prompts. Can SAM also achieve the same effect? What is the significance of studying the specific RRIS model?

**Suitability:**

3

---

### Official Review · Reviewer_sCDn · 2024-05-20

**Rating:** 4
**Confidence:** 4

**Summary:**

The authors rethink the issues existing in implicit optimization paradigms based on the traditional RRSIS methods, and propose solutions from the perspective of dual alignment. The authors designed two explicit alignment modules, namely explicit alignment strategy and reliable proxy alignment module. This method has shown significant advantages in both theory and practice, with high innovation and practical value. The effectiveness and robustness of the proposed method have been validated through extensive experiments conducted on two different RRSIS datasets. The experimental results show that the proposed DANet significantly improves segmentation performance without adding additional learnable parameters, outperforming existing state-of-the-art methods.

**Strengths:**

1. The explicit alignment strategy proposed in the paper effectively reduces the difference between visual and text domains by reducing the cross domain affinity distribution. This method achieves consistency between visual features and text activation through affinity learning, avoiding the problem of fuzzy fusion in traditional methods. The explicit alignment strategy not only improves the model's cross domain alignment ability, but also enhances its processing ability for complex semantic scenes.

2. The reliable proxy alignment module aims to improve the multimodal perception ability of the predictor and reduce the interference of deceptive noise. By adaptively selecting reliable proxies in visual features, this module can better capture and utilize semantic information in remote perception scenes, avoiding uncertainty interference caused by direct matching. This module enhances the robustness of the model through reliable matching, making it perform better when dealing with complex visual cues.

3. The paper conducted extensive experiments on two different RRSIS datasets, and the experimental results proved the effectiveness and superiority of the proposed method. Compared with state-of-the-art methods, DANet performs well in multiple benchmark tests, significantly improving segmentation accuracy. These experimental results strongly support the theoretical assumptions and method design of the paper.

**Limitations:**

1. There are some typos in the paper, such as "works have have turned" on page 1, line 67. It is recommended to carefully check and correct them.

2. There is ambiguity in Figure 1 (c). Is the Text Encoder in the Fixed Kernel-based Predictor? This may lead to confusion in readers' understanding of the method. It is recommended to provide detailed explanations or modifications to the illustrations to improve their clarity and accuracy.

3. The paper lacks a time complexity analysis of the proposed method. It is recommended to supplement time complexity analysis to comprehensively evaluate the efficiency and practical application value of the method.

**Suitability:**

3

---

### Official Review · Reviewer_3rha · 2024-05-21

**Rating:** 4
**Confidence:** 3

**Summary:**

This paper discusses the two problems of the existing RRSIS method based on implicit optimization, namely, the misalignment between domains in the encoding stage and the semantic agnostic prediction in the decoding stage, and gives their own explicit alignment methods respectively, and combines them to form the dual alignment network proposed in this paper. The method proposed in this paper improves the segmentation effect of the model by aligning twice in the encoding and decoding stages, and verifies the accuracy of the model on two public datasets.

**Strengths:**

1. By analyzing the problems existing in the existing implicit optimization strategy, a new dual alignment network is designed to solve the two problems of non-aligned encoding and unknown semantic segmentation from the perspective of explicit alignment.
2. A RAA module is proposed to select more accurate regions by comparing the distribution similarity between a group of pixels and their corresponding semantics, which effectively reduces the mis-segmentation caused by the deceptive noise generated by the long-distance imaging mode in the aerial image and improves the accuracy of segmentation.
3. Compared with the implicit optimization method, the explicit optimization strategy proposed in this paper achieves performance improvement without the need for additional learning parameters, and also improves the interpretability of the model.
4. A wide range of experimental results are provided, and the actual effect of the proposed method is verified on multiple benchmarks.

**Limitations:**

1. Although the paper mentions that no additional parameters are added, the dual alignment process may still require more computational consumption, and the paper lacks relevant experimental tables.
2. The paper lacks verification of the robustness of the dual alignment strategy. The RRSIS-D data includes 15,000+ training data. If the amount of data is reduced, will the change in model performance decrease significantly?
3. The paper lacks discussion of the defects of the proposed method.

**Suitability:**

2

---

### Official Review · Reviewer_t34j · 2024-05-24

**Rating:** 3
**Confidence:** 3

**Summary:**

The task of Referring Remote Sensing Image Segmentation (RRSIS) involves identifying specific regions in aerial images that match given textual descriptions. Current methods often use implicit optimization, which includes early cross-modal feature fusion and a fixed convolutional kernel-based predictor. These methods neglect the inter-domain gap and perform class-agnostic predictions. In this manuscript, the authors propose a novel approach from a dual-alignment perspective to address these issues. The proposed DANet includes an explicit alignment strategy and a reliable agent alignment module. Extensive experiments on two remote sensing datasets show that DANet achieves superior segmentation performance compared with the SOTA methods.

**Strengths:**

1. The proposed explicit alignment strategy could effectively reduce domain discrepancies and the reliable agent alignment module could enhance multi-modality awareness and mitigate noise interference.
2. Extensive experiments on the RefSegRS and RRSIS-D datasets demonstrate that the proposed methods could effectively boost the performance.

**Limitations:**

1. The motivation shown in Figure 1 is not very clear, as it seems only show the difference between the proposed method and the existing method.
2. In addition to the current metrics (mIoU, oIoU, and Pr@X), inference efficiency, parameters, and computational complexity are also important and need to be further compared with comparison methods. The proposed reliable agent alignment module increases the size and the complexity of the model.
3. What are the advantages of the proposed DANet compared with the currently popular pre-training large multimodal models?
4. The performance of this work is limited and lacks comparison with the latest state-of-the-art methods.
5. The authors choose BERT as their text encoder. However, why not employ a more recent contextualized representation for the text encoder? A transformer-based representation might work better, such as GPT-2 and T5, or a multi-modal text encoder, such as CLIP text encoder.
6. Compared with the current large multimodal model, the value and significance of this work is not high enough. Authors should choose to proceed on a large multimodal model.

**Suitability:**

3

---

### Meta-Review · Area_Chair_nXei · 2024-06-29

**Recommendation:** Accept (Poster)
**Confidence:** 5

**Metareview:**

Referring Remote Sensing Image Segmentation (RRSIS) is a very novel and highly valuable research area. After one round of rebuttal from the authors, all reviewers have ultimately raised their scores to positive. Therefore, I believe this paper can be accepted as a poster.